# Teacher Trainees’ Well-Being—The Role of Personal Resources

**DOI:** 10.3390/ijerph19148821

**Published:** 2022-07-20

**Authors:** Elena Hohensee, Kira Elena Weber

**Affiliations:** 1Zukunftszentrum Lehrkräftebildung: ZZL-Netzwerk, Leuphana University Lüneburg, Universitätsallee 1, 21335 Lüneburg, Germany; 2Department of Educational Research and Educational Psychology, IPN—Leibniz Institute for Science and Mathematics Education, Olshausenstraße 62, 24118 Kiel, Germany; k.weber@leibniz-ipn.de

**Keywords:** teacher trainees, well-being, health literacy, occupational self-regulation, teachers’ health promotion

## Abstract

Teacher well-being is intrinsically associated with their personal resources, including health literacy and occupational self-regulation. However, there are few empirical findings on teacher trainees’ health literacy. Furthermore, occupational self-regulation has so far only been associated with indicators of occupational well-being. From a public health perspective, research on teacher trainees’ general well-being will benefit from taking both research aspects into account. In this study, we analysed data from 407 teacher trainees in Germany. Latent profile analysis confirmed the four occupational self-regulatory types (healthy-ambitious, unambitious, excessively ambitious, and resigned), which differed significantly on the health literacy dimensions self-regulation, self-control, self-perception, proactive approach to health, communication and cooperation, and dealing with health information. The health literacy dimensions of self-regulation and self-control were mainly related to occupational self-regulation. Independently of each other, the self-regulatory types and the health literacy dimensions of self-regulation, self-control, and proactive approach to health predicted teacher trainees’ general well-being. If both constructs are considered together, the health literacy dimensions explain more variance in teacher trainees’ general well-being than the self-regulatory types. Research and practical implications are discussed.

## 1. Introduction

The teaching profession is associated with various work-related stressors (e.g., heavy workloads, time pressures, and a variety of social interactions almost every day) and is considered one of the most stressful occupations [1,2,3,4]. This is especially true for teacher trainees, who are confronted day after day with their unique challenges (e.g., Kiel and Weiß [5]). During the last decades, the well-being of teachers has gained increasing attention [6,7]. In their recent review, Hascher and Waber [6] underlined the fact that high teacher *well-being* is positively associated with positive emotions and satisfaction and negatively associated with emotional exhaustion and burn-out. In accordance with the salutogenic approach, well-being is not just the opposite of stress but a multidimensional construct, and can be distinguished into *general well-being*, defined as ‘open, engaged, and healthy functioning’ [8] (p. 47), or *occupational well-being*, which refers to healthy functioning in the work environment [9]. Regarding the relationship between these two categories of well-being, teachers who report high *general well-being* also tend to report high *occupational well-being* [10,11]. Accordingly, we advocate a public health approach that explores teachers’ *general well-being* across multiple areas of life (including work and working conditions); thus, we focus on teachers’ *general well-being* in our study.

To maintain and foster *general well-being*, the personal resources of teachers are of high importance (e.g., [12,13]). One personal resource used to meet occupational demands is *occupational self-regulation* [14], which can be described as the responsible use of one’s own resources [15]. *Occupational self-regulation* has been widely researched in the context of teachers’ *occupational well-being* [16], as it influences job satisfaction and the maintenance of professional productivity and effectiveness [17]. It is also seen as an aspect of teachers’ *occupational health literacy* [18] (p. 21).

A newer construct in the context of teachers’ *well-being* is *health literacy*. Health literacy can be understood as a personal health-related resource that is primarily determined by perceptive-motivational dimensions (*self-perception* and a *proactive approach to health*) and behavioural dimensions (*dealing with health information*, *self-control*, *self-regulation*, and *communication and cooperation*) and can make a vital contribution to the maintenance of teachers’ *well-being* [19,20].

Overall, both resources seem to be important for teachers’ *well-being*, but to our knowledge, there is an absence of studies which simultaneously investigated both resources. We argue that these two personal resources should not be considered separately, and that research in the context of teacher trainees’ *health literacy* and *well-being* will profit from taking both aspects into account. Therefore, in the present study, we investigated not only how these two personal resources are developed among teacher trainees but also how these resources are related to each other. Further, the association with teacher trainees’ general *well-being* and the relative importance of *health literacy* and *occupational self-regulation* in predicting *well-being* are emphasised. To this end, we reveal a more comprehensive picture of how a work-related personal resource (*occupational self-regulation*) and a health-related personal resource (*health literacy*) predict teacher trainees’ *well-being* from a public health perspective.

## 2. Teachers’ Well-Being

Promoting and maintaining teachers’ well-being is an important prerequisite for teaching quality and student achievement [16,21]. Teachers who suffer from impaired health report reduced job satisfaction and are therefore more likely to leave the teaching profession [22], whereas high teacher well-being is positively associated with resilience, motivation, and commitment (e.g., [23,24]). Teachers’ well-being has mostly been examined through a focus on negative dimensions of well-being, such as stress and burnout [25]. However, during the last decade, the focus of research has shifted towards a more positive and resource-orientated perspective. This shift is important to avoid a deficit view [26] and to examine personal resources [27]. In this study, we focus on two personal resources and their importance for teachers’ *general well-being*: *health literacy* and *occupational self-regulation*.

### 2.1. Teachers’ Well-Being and the Importance of Health Literacy

One multifaceted construct associated with *well-being* is *health literacy*, which can be seen as a personal resource for teachers’ *well-being*. *Health literacy* is considered a determinant of health [28] and is integrated into various World Health Organization initiatives, such as the Shanghai Declaration [29], the Roadmap for the Promotion of Health Literacy over the Life Course [30], and the Manifesto on Health Literacy [28]. Limited *health literacy* is associated with lower physical and psychological *well-being* [31] and lower self-reported health status [32].

Several studies have investigated the *health literacy* of (prospective) teachers. For example, pre-service teachers in Tehran rated their *health literacy* as inadequate to problematic [33]. Another cross-sectional study among secondary school teachers from Sri Lanka identified limited health literacy among 32.5% of the study sample (*n* = 520) [34]. Empirical research on primary and secondary school teachers in Germany has indicated that more than half of them have limited *health literacy* [35]. In the area of health promotion, teachers find it particularly difficult to deal with mental health issues and have great difficulty finding information to improve their mental well-being [35]. Previous results have tended to indicate a low level of *health literacy* among prospective teachers [33,34,35], but research in this field is still limited and inconsistent.

Lenartz ([20] (p. 138), German version) and Soellner et al. ([36] (p. 251), English version) developed a structural model of *health literacy* from a health promotion perspective, which provides the theoretical and analytical framework in this study. Health literacy in this model is understood as a kind of personal health-related resource and encompasses the totality of skills and abilities that a person must have to act in everyday life and deal with the health system in a way that has a positive impact on their health and well-being [37]. In addition to health-related knowledge and basic health-related skills, the core of the model consists of perceptive-motivational dimensions (*self-perception* and *proactive approach to health*) and behavioural dimensions (*dealing with health information, self-control, self-regulation,* and *communication and cooperation*) (for a figure of the structural model of health literacy, see [20,36]).

In this study, we focus only on these core dimensions of *health literacy*. Recent findings in the context of teacher education indicate that student teachers rate their *health literacy* predominantly better than teacher trainees [38]. Student teachers rated themselves significantly better in all dimensions, except *communication and cooperation*. For both groups, the highest means were found in the dimension *dealing with health information* and the lowest in *self-regulation* [38].

Lenartz [20] identified that up to 42% of the variance in mental health could be explained by the dimensions of *self-perception*, *proactive approach to health*, *self-regulation*, and *self-control* in particular. Furthermore, 40% of the variance in the prevalence of health complaints could also be explained by these dimensions [39]. Other studies have emphasised that both *self-regulation* and higher means of *self-perception* were significantly associated with managers’ general well-being, the former directly and the latter indirectly [40]. Overall, the components of *health literacy* contained in Lenartz’s structural model describe the central prerequisites for health-promoting behaviour and one’s own health. However, research in the field of teacher education is still rare, and it remains unclear how the work-related personal resource *occupational self-regulation* is associated with the *health literacy* of teacher trainees.

### 2.2. Teachers’ Well-Being and the Importance of Occupational Self-Regulation

In the prominent model of teachers’ professional competence, in addition to professional knowledge and beliefs, motivation and *occupational self-regulation* are seen as important prerequisites for successfully coping with professional demands [41]. Occupational self-regulation is described as the responsible use of one‘s own resources [15]. It represents a resource for teachers’ *occupational well-being* [16] and is central to job satisfaction and the maintenance of professional productivity and effectiveness [17]. Klusmann et al. [16] understood adaptive self-regulation as a combination of high occupational engagement and high occupational resilience [42], which is also adopted in the present article. Occupational engagement is understood as a basic willingness to invest resources (i.e., energy and effort) in one’s profession. Occupational resilience is described as the ability to distance oneself from occupational concerns and to cope successfully with failure, that is, to protect resources ([14,43]). ‘It includes emotional distancing, a low tendency to give up after failure, active coping, and mental stability’ [16] (p. 704). The intra-individual interaction of the two factors can be operationalised by the multidimensional personality diagnostic procedure described by Schaarschmidt and Fischer [43,44], which identifies work-related behaviour patterns (AVEM). The procedure records health-promoting or health-threatening behaviours in coping with occupational demands and integrates these dimensions into four self-regulatory patterns (the healthy-ambitious, unambitious, ambitious, and resigned types). The healthy-ambitious type (H) is characterised by a high level of occupational engagement and pronounced resistance to stress and is the most adaptive self-regulatory pattern. The unambitious type (U) is characterised by low occupational engagement but high resilience, whereas the excessively ambitious type (A) scores high on engagement and low on resilience. A low level of engagement with work and a low level of occupational resilience describe the resigned type (R) [45]. The last two types are thought to be at high risk for burnout and stress [16,43].

Previous empirical findings have confirmed these four types [16,17,46,47]. A first study by Klusmann et al. [16] identified that 54.6% of the participating teachers tended to classify themselves as a health-promoting pattern (healthy-ambitious and unambitious types). Recent research on teacher trainees [47,48] indicated that slightly less than half of them assign themselves to health-threatening patterns (ambitious and resigned types) [47,48]. Klusmann et al. [16] found correlations between affiliation to the types and *occupational well-being* in terms of emotional exhaustion and job satisfaction. Teachers of the healthy-ambitious type reported the most favourable results for job satisfaction, followed by teachers of the unambitious type. Moreover, teachers of the healthy-ambitious and unambitious types reported less emotional exhaustion compared to teachers of the ambitious and resigned types [8]. To our knowledge, associations of these self-regulatory patterns with teacher trainees’ *general well-being* have not been investigated thus far, and it remains unclear how *occupational self-regulation* is associated with teacher trainees’ *health literacy*.

## 3. Research Questions and Hypotheses

Teacher trainees’ *general well-being* could be fostered by personal resources, such as *occupational self-regulation* and *health literacy*. To our knowledge, no studies have investigated the association between the work-related personal resource *occupational self-regulation* and the health-related personal resource *health literacy*. Our research aims to narrow this research gap and to shed more light on the field of *teacher trainees*’ *health literacy* and *teacher trainees’ general well-being*. The following research questions are addressed:How developed are teacher trainees’ personal resources of *health literacy* and *occupational self-regulation*?

**H1.** 
*To our knowledge, there are only a few studies in the field of health literacy that have focused on (prospective) teachers (e.g., [35]). In a recent study [46], we showed that student teachers possess better health literacy than teacher trainees. Therefore, we assume that teacher trainees’ health literacy, especially the dimensions of self-regulation and communication and cooperation, are not well developed.*


**H2.** 
*There is considerably more research on occupational self–regulation of teacher trainees, and several findings (e.g., [47,48]) indicate that most teacher trainees can be characterised as health-promoting types. In line with this, we hypothesise that the majority of teacher trainees in our study can also be assigned to the health-promoting types. However, we conducted our research during the COVID-19 pandemic; therefore, the results may be slightly different to those of previous studies.*


2.How is occupational self-regulation related to teacher trainees’ health literacy?

**H3.** 
*We assume that teacher trainees who belong to the healthy-ambitious type (H type) also have the highest health literacy. Regarding the different dimensions of health literacy, we assume that self-regulation and self-control have the highest associations with occupational self-regulation because of similar psychological terminology.*


3a.How are the two personal resources related to teacher trainees’ *general well-being*?

**H4.** 
*We assume that the correlation between well-being and the dimensions of health literacy is higher than the correlation between the dimensions of occupational self-regulation. A study on managers showed that the health literacy dimensions of self-regulation and self-control were the strongest predictors of general well-being [40]. Due to the lack of studies on teachers, we explored how the dimensions of teachers’ occupational self-regulation are associated with teacher trainees’ general well-being.*


3b.What is the relative importance of *health literacy* and *occupational self-regulation* in predicting teacher trainees’ *general well-being,* in terms of the portion of variance?

**H5.** 
*Health literacy is expected to explain a higher specific portion of variance in well-being than occupational self-regulation because it is a health-related personal resource rather than a work-related personal resource and therefore should have greater influence on one’s general well-being. Moreover, previous findings in other research areas [20,40] have shown that the health literacy dimensions of self-regulation and self-control are the most important predictors for well-being. Regarding occupational self-regulation, we assume that the health-promoting types in particular explain a portion of variance of well-being [16]. The same was expected for the shared portion of variance explained by both predictors.*


## 4. Methods

### 4.1. Participants

Teacher education in Germany is divided into three phases (Phase 1: student or pre-service teachers; Phase 2: teacher trainees; and Phase 3: in-service teachers). The first phase takes place at the university and usually lasts five years. Student teachers receive theoretical and empirical knowledge and gain their first teaching experience. Following this, the traineeship, also known as the referendariat, takes place, which is intended to provide more practical knowledge and usually lasts between 18 and 24 months. During the traineeship, teacher trainees work in the school but receive support and feedback from mentors and teacher training colleagues and are evaluated by them. After this period, teacher trainees develop into in-service teachers and work full time while they continue their professional development (for more details, see [49]). Our sample consisted of 407 teacher trainees in Germany (female: 82.6%) recruited from eight teacher training colleges (One part of this sample was used in a previous study [38]). The sample taught at primary schools (i.e., Grades 1–4; 50.4%) and secondary schools (i.e., Grades 5–13; 49.6%). The teachers in our sample reported a mean age of 28.3 years (SD = 5.0).

### 4.2. Instruments

*Occupational Self-regulation.* Occupational self-regulation consists of occupational engagement and resilience [16], which were measured using eight subscales from the occupational stress and coping inventory (AVEM; [43]). Participants responded on a five-point scale ranging from 1 = strongly disagree to 5 = strongly agree. *Occupational engagement* was measured using the following subscales: significance of work, career ambitions, exertion, and perfectionism. *Occupational resilience* was measured with four subscales: emotional distancing, low tendency to give up, active coping, and mental stability.

*Health Literacy.* Health literacy was measured based on the German questionnaire by Lenartz ([50]; validated by Lenartz [20] and Soellner et al. [36]), which has already been used in other studies (e.g., [39,51]). The questionnaire focused on the following subscales: *self-regulation, self-control, self-perception, proactive approach to health, communication and cooperation*, and *dealing with health information*. All items were rated on a four-point Likert scale ranging from 1 = strongly disagree at all to 4 = strongly agree.

*Well-being.* The WHO-5 Well-being Index (1998 version) is a short, self-report questionnaire containing five positively worded items related to positive mood, vitality, and general interests. It is recommended and successfully used for screening depressive disorders [52], but it is also used in various studies regarding teachers’ well-being (see the review of [6]). Respondents were given five statements and were asked to indicate how they had been feeling over the past two weeks using a six-point Likert scale (0 = none of the time to 5 = all of the time). Responses were summed so that the final total score ranged from 0 (absence of well-being) to 25 (maximal well-being). A score below 13 indicates low mental well-being and a risk of depression.

### 4.3. Procedures

Participants were recruited via their teacher training colleges. The survey took place online in the context of a pedagogical seminar; only one college sent the study description and the link to the survey (LimeSurvey) by mail to the teacher trainees. All teachers completed the online questionnaire and answered demographic items (e.g., gender, age) and the self-report measures outlined above. The data collection was initiated at the end of November 2020 and lasted 12 weeks.

Regarding ethical approval of the study, the German Research Foundation (DFG) states that a study requires ethical approval whenever the participants, for example, have to endure high emotional or physical strains, cannot be fully informed about the purpose of the study, are patients, or undergo functional magnetic resonance imaging or transcranial magnetic stimulation during the course of the study. Our study did not affect any of the above-mentioned conditions and therefore did not require ethical approval. However, the initial consent page informed teachers about the study’s purpose and confidentiality as well as data protection information, and teachers were informed that they could drop out at any time.

### 4.4. Statistical Analyses

Preliminary analyses were conducted with the aim of reporting descriptive statistics. The determination of *occupational self-regulation* followed a person-centred approach and was described via four self-regulatory patterns (healthy-ambitious type (H), unambitious type (U), excessively ambitious type (A), and resigned type (R) [16,44,47]), which were conducted by latent profile analysis (LPA). We used four criteria to identify the number of latent clusters [17]. First, adding an additional class should lead to a decrease in the Akaike information criterion (AIC), Bayes information criterion (BIC), and sample-size-adjusted BIC (SSA-BIC). Second, the Lo–Mendell–Rubin likelihood ratio test (LMR test; [53]) gives a p-value that tests the null hypothesis that a solution with *k* classes has the same goodness of fit as a solution with *k*-1 classes. Third, entropy can range from 0 to 1, with higher values representing a better fit of the profiles to the data and values of 0.80 or greater providing supporting evidence that profile classification of individuals in the model occurs with minimal uncertainty [54,55]. Lastly, the interpretability of the different cluster solutions was assessed against the background of their theoretical plausibility [56,57] and previous empirical findings on behavioural styles. The LPA was conducted with MPlus software [58].

Intercorrelations and correlations were calculated to highlight the relation within the personal resources, as well as with general well-being. We also performed analyses of variance (ANOVA) with post-hoc tests (Tukey) to compare mean health literacy across self-regulatory patterns. A Bonferroni-type adjustment was made to avoid inflated Type I errors through multiple testing. With a familywise error rate (α) of 0.05 and six significance tests, the critical value for each test (α) was adjusted to 0.0086.

We designed multiple regression models to analyse the relative importance of health literacy and occupational self-regulation in predicting teacher trainees’ well-being. Separate multiple-regression analyses were conducted to highlight the association between both personal resources and general well-being (M1–3).

## 5. Results

### 5.1. Well-Being and Personal Resources

Using the scores of the WHO-5, 40.3% (*n* = 164) of the participants were screened as having a low *general well-being* (score < 13). Overall, the average mean of 13.13 (SD = 5.00) was on the threshold of low *well-being*.

The first research question was related to the state of teacher trainees’ personal resources: *health literacy and occupational self-regulation*. The means, standard deviations, and internal consistencies are provided in Table 1.

Table 2 shows the results of the latent profile analysis to identify the self-regulatory patterns, starting with a one-to-five-class solution based on the z-standardised mean values of the eight subscales of engagement and resilience. The fit indices supported the three-class solution, but based on theoretical assumptions and empirical evidence on latent classes of the AVEM (e.g., [16]), we decided to take the data as confirmation of the four-class solution for teacher trainees. The z-scores of the eight scales also supported the four-class solution and showed the respective differences between the resilience and engagement scales (Figure 1).

Class 1 reflected the H type (35.4%) teacher trainees who had above-average scores in engagement and resilience compared to the mean of the overall sample. Class 2, which corresponded to the pattern of the U type (22.9%) teacher trainees, was characterised by very low scores on the engagement and high scores on the resilience scales. Class 3, representing the risk A type (19.2%) teacher trainees, was characterised by very high means of engagement and predominantly low means of resilience. Class 4, which had below-average scores in both the engagement and resilience scales compared to the overall sample, represented the R type (22.6%) teacher trainees.

Referring to teacher trainees’ *health literacy*, they rated themselves highest in the dimensions *dealing with health information* (M = 3.21, SD = 0.51) and *self-perception* (M = 3.16, SD = 0.47), and lowest in *self-regulation* (M = 2.45, SD = 0.56).

### 5.2. Relationship between Health Literacy and Occupational Self-Regulation

Research Question 2 focused on the relationship between teacher trainees’ health literacy and self-regulatory patterns. The results of the correlation analyses indicated how the *occupational self-regulation* dimensions and *health literacy* dimensions were related to each other (see Table 3). Strong and medium correlations (r > 0.30) between the personal resources were found between the *health literacy* dimensions *self-regulation* and *self-control* and the subdimensions of *occupational self-regulation*. For the *health literacy* dimension *self-regulation*, a strong positive correlation was found with the subdimension *emotional distancing* (r = 0.67), medium positive correlations with the dimensions *low tendency to give up* (r = 0.46) and *mental stability* (r = 0.44), and a medium negative correlation with the dimension *exertion* (r = −0.43). Medium positive correlations were found between the dimensions of *career ambitions* (r = 0.30), *p**erfectionism* (r = 0.34), and *active coping* (r = 0.36), and the *health literacy* dimension of *self-control*.

The results of the ANOVA are shown in Table 4. Substantial differences in the *health literacy* dimension of *occupational self-regulation* were reported by the four self-regulatory patterns (F (3,403) = 48.75, *p* > 0.001, η^2^ = 0.27). The effect size was large and indicated that the patterns accounted for about 27% of the variance in the *health literacy* dimension of *self-regulation*. As post-hoc analyses (Tukey) showed, teacher trainees belonging to the H type and U type reported statistically better *self-regulation* than teacher trainees of the A and R types. In terms of the effect sizes, the differences between the average *self-regulation* of the H type and both the R type (Cohen’s *d* = 1.33) and A type (Cohen’s *d* = 1.15) teacher trainees can be classified as large. Large standardised differences were also observable between the U type and the R type (Cohen’s *d* = 1.22) and between the U type and the A type (Cohen’s *d* = 1.05) teacher trainees.

Substantial differences were also indicated for *self-control* reported by the four self-regulatory patterns (F (3,403) = 23.30, *p* > 0.001, η^2^ = 0.15). The effect size was large, indicating that the patterns explained about 15% of the variance in *self-control*. Post-hoc analyses (Tukey) showed that the H type teacher trainees scored the highest on *self-control*, followed by the A type. Teacher trainees of the U type and R type scored the lowest and did not differ from each other. In terms of the effect sizes, the difference between the *self-control* of the H type and U type (Cohen’s *d* = 0.94) teacher trainees can be classified as large, whereas a medium-sized difference was observed between the H type and R type (Cohen’s *d* = 0.76) teacher trainees. Similar results were found for the A type teacher trainees. A large standardised difference was observed between the A type and U type (Cohen’s *d* = 0.86) teacher trainees, and a medium-sized difference was observed between the A type and R type (Cohen’s *d* = 0.68) teacher trainees.

Substantial differences in *self-perception* were reported by the four self-regulatory patterns (F (3,403) = 5.31, p < 0.001, η^2^ = 0.04). The effect size was medium and indicated that the patterns accounted for about 4% of the variance in *self-perception*. As post-hoc analyses (Tukey) showed, teacher trainees belonging to the H type reported statistically better *self-perception* than teacher trainees of the R type. In terms of the effect sizes, the differences between the average *self-perception* of the H type and the R type (Cohen’s *d* = 0.55) teacher trainees were classified as medium.

Similar results were found for *proactive approach to health* and *dealing with health information* of the four teacher trainee types. Overall, we found substantial differences in *proactive approach to health* reported by the four types (F (3,403) = 5.85, *p* < 0.001). The moderate effect size (η^2^ = 0.04) indicated that about 9% of the variance in *proactive approach to health* could be explained by the self-regulatory patterns. Post-hoc analyses (Tukey) showed that the H type teacher trainees scored the highest on *proactive approach to health*, and the R type teacher trainees scored the lowest. In terms of the effect sizes, the differences between the average *proactive approach to health* of the H type and the R type teacher trainees were classified as small (Cohen’s *d* = 0.49).

Substantial differences were also indicated for *dealing with health information* reported by the four self-regulatory patterns (F (3,403) = 5.14, *p* < 0.001, η^2^ = 0.04). The effect size was medium and indicated that the patterns accounted for about 4% of the variance in *dealing with health information*. As post-hoc analyses (Tukey) showed, teacher trainees belonging to the H type reported statistically better *dealing with health information* than teacher trainees of the R type. In terms of the effect sizes, the differences between the average *dealing with health information* of the H type and the R type teacher trainees were classified as medium (Cohen’s *d* = 0.51).

Substantial differences in *communication and cooperation* were reported by the four self-regulatory patterns (F (3,403) = 8.74, *p* > 0.001, η^2^ = 0.06). The effect size was medium and indicated that the patterns accounted for about 6% of the variance in *communication and cooperation*. On average, teacher trainees of the U type reported a better *communication and cooperation* than the H type, but they did not differ significantly from each other. As post-hoc analyses (Tukey) showed, teacher trainees belonging to the U type and H type reported better *communication and cooperation* than teacher trainees of the A and R types. In terms of the effect sizes, the differences between the average *communication and cooperation* between the H type and the A type (Cohen’s *d* = 0.47) or R type (Cohen’s *d* = 0.45) teacher trainees were classified as small. Medium standardised differences were observable between the U type and the A type (Cohen’s *d* = 0.60) and between the U type and the R type (Cohen’s *d* = 0.58) teacher trainees.

### 5.3. Associations between the Two Personal Resources and Well-Being

To answer research question three, Table 3 shows the correlations between *occupational self-regulation* dimensions and *health literacy* dimensions and *general well-being*. *Well-being* correlated predominantly with the subdimension *occupational resilience* (emotional distancing, low tendency to give up, active coping, and mental stability) of *occupational self-regulation*, with the highest correlation being with *emotional distancing* (r = 0.41). The highest correlations between the dimensions of *health literacy* and *well-being* were found for the *self-regulation* dimension (r = 0.47). Medium correlations were also found for the *proactive approach to health* dimension (r = 0.35).

We hypothesised that teacher trainees’ *general well-being* was associated with their *self-regulatory patterns* and *health literacy*. Three multiple regression analyses were conducted (see Table 5). The categorical variable self-regulatory pattern was dummy coded, and assignment to the R type was taken as the reference category. Hence, the regression coefficients of the self-regulatory pattern must be interpreted relative to teacher trainees of the R type. The results revealed a statistically significant association between teacher trainees of the H, U, and A types and *well-being* (M1). Teacher trainees of these types reported better *general well-being* compared to teacher trainees of the R type. However, the amount of variance explained in *well-being* was small (R^2^ = 0.09). The results of the second regression model (M2) with the health literacy dimensions as predictors identified that *proactive approach to health* (β = 21, *p* < 0.001), *self-control* (β = 15, *p* < 0.001), and *self-regulation* (β = 39, *p* < 0.001) had a positive, significant association with teacher trainees’ *general well-being*. The variance explanation of teacher trainees’ *well-being* was 28% and corresponded to strong variance elucidation ([59] see Table 5).

If both resources for the elucidation of *general well-being* were considered, similar results were found for *health literacy* (*self-control*: β = 0.10, *p* < 0.05; *proactive approach to health*: β = 0.21, *p* < 0.001; *self-regulation*: β = 0.38, *p* < 0.001). Regarding the self-regulatory types, only teacher trainees of the H type (β = 14, *p* < 0.05) still had a positive, significant association with *well-being*, but the β-coefficient decreased (see M3). The amount of variance explained in *general well-being* was strong (R^2^ = 0.29).

## 6. Discussion

In the present study, we aimed to shed more light on both teacher trainees’ personal resources in terms of *occupational self-regulation* and *health literacy*. We argue that both constructs are highly important concerning the *general well-being* of teachers and that they should be considered collectively in health-related research. We initially investigated how these personal resources were developed among teacher trainees and then further investigated how *occupational self-regulation* was related to teacher trainees’ *health literacy*. Moreover, we analysed associations with teacher trainees’ *general well-being*.

Regarding Research Question 1, the results of our study revealed that 35.4% of the teacher trainees associated themselves with the H type, 22.9% with the U type, 19.2% with the A type, and 22.6% with the R type. Thus, Hypothesis 2 was confirmed. Similar percentage distributions of the self-regulatory patterns for teacher trainees appeared in previous studies (e.g., [17,47]). Compared to the teacher sample in Klusmann et al.’s [16] study, our sample had a higher proportion of teachers with patterns regarding *health-promoting types*. Nevertheless, 41.8% classified themselves into a health-threatening pattern, which was associated with lower *occupational well-being*, reflected by higher emotional exhaustion and lower job satisfaction than the H type [16]. Overall, the results underline the need for learning opportunities in the context of teacher education to develop the acquisition of *occupational self-regulation* [17,60]. This was also confirmed by Lohse-Bossenz and Rutsch’s [47] study results. They investigated whether these self-regulatory patterns changed during the referendariat and identified a pattern change in approximately 44% of the teacher trainees, which illustrated the dynamics of professional experience and behaviour [47].

Regarding our results for teacher trainees’ *health literacy*, the findings revealed the highest scores in the dimension *dealing with health information* and the lowest in *self-regulation*, which has also been presented in previous findings of student teachers and teacher trainees [38,61]. Thus, Hypothesis 1 was confirmed. Overall, the behaviourally relevant components (*self-control*, but especially *self-regulation*), which refer to the inner processes and procedures that enable a person to act and implement the intended actions [20], are not well developed according to the self-assessments of the teacher trainees in our sample. Accordingly, our results draw attention to the importance of *health literacy* in the context of teacher education [62,63]. Regarding the structural model of health literacy, these findings necessitate promotion measures and follow up on Lamanauskaus’ claim that health-related competences have been neglected in teacher education [64]. Its early promotion has been widely discussed in the school setting [65,66], but (trainee) teachers should have well-developed *health literacy* as health promoters [67,68]. It is considered a necessary prerequisite for health-promoting behaviour and the maintenance and promotion of health [19].

However, we acknowledged that our study was conducted during the COVID-19 pandemic; therefore, the extent to which contextual factors influenced teacher trainees’ *health literacy* and their *occupational self-regulation* remains unclear. Regarding *health literacy*, recent findings from Germany [69] revealed that the majority of the population is able to find information about activities that are good for mental health and *well-being* [69]. Our results for teacher trainees are somewhat similar, with the highest scores recorded for *dealing with health information*. However, the authors highlighted the fact that there are still difficulties in finding information for health promotion in one’s own environment (e.g., workplace, school) [69]. Occupational stressors could also have an influence on teacher trainees’ *health literacy* and *occupational self-regulation*. The second phase of teacher education (referendariat) is often experienced by teacher trainees as a phase of adjustments, stresses, ambivalences, and has the potential for conflict [70]. This is primarily due to various stressors, such as the perceived high workload, the pressure to perform, conflicts with students, or the perceived stressful dependence on instructors (e.g., [48,71,72,73,74]). How these occupational stressors are related to *health literacy* is an open question and should be investigated in more detail in further studies.

Our study also offered new insights into teacher trainees’ *health literacy* research by exploring the association between *health literacy* and *occupational self-regulation* (Research Question 2). In relation to the *health literacy* of teacher trainees, 27% of the variance in *self-regulation* and 15% in *self-control* could be explained by occupational self-regulatory patterns, whereby the post-hoc results revealed that the H type teacher trainees differed most significantly from the R type teacher trainees. Teacher trainees of the H type had the highest scores in all dimensions of *health literacy*. The lowest means for the *health literacy* dimension *self-regulation* were found on the health-threatening pattern (A and R types), which also revealed low occupational resilience. The highest means of *self-control* were indicated among the types that showed high occupational engagement (H and A types). The correlations between the two personal resources provide a possible explanation. In particular, the individual dimensions of *occupational engagement* and *resilience* were related to the *health literacy* dimensions of *self-regulation* and *self-control*. Thus, Hypothesis 3 was confirmed. The strongest correlations between the *health literacy* dimension of *self-regulation* and the dimensions of *occupational self-regulation* revealed, for example, in the dimensions of *emotional distancing, low tendency to give up,* and *mental stability*. In terms of content, this can also be found in the description by Lenartz [20]: *self-regulation* comprises skills and abilities in dealing with tension and stress, the reduction of tension, the ability to relax, and the change between concentration and relaxation [20] (p. 135). Similar findings were revealed for the *health literacy* dimension of *self-control*, which was mainly related to the subdimensions of *occupational self-regulation*: *active coping, perfectionism*, and *career ambitions*. *Self-control*, for example, aims to implement and enforce behaviour once it has been decided, plans are kept in mind, planned actions are implemented with discipline, and distractions and digressive thoughts are overcome [20] (p. 135). This suggests that the development of *occupational self-regulation* strategies could be derived concerning promoting the *health literacy* of teacher trainees. Teachers need self-regulatory skills that enable them to use action- and emotion-related strategies in ways that are functional for coping with job demands and health issues [75]. Mindfulness represents a possible emotion-related form of coping and can help identify and regulate individual stress patterns [76], as well as promote self-care and overall *well-being* [77]. The relevance of mindfulness in the educational context has been investigated in numerous studies, and the positive effects of mindfulness-based interventions in relation to novice teachers and teachers seem to be promising (see recent meta-analyses [78,79,80]). The usefulness of *self-regulatory* strategies has been highlighted both for the teaching profession in general and for beginning teachers in particular [81].

Regarding teacher trainees’ *general well-being*, although 59.7% rated their *well-being* as sufficient, the mean score was on the threshold of low well-being. One explanation for this could be derived from previous findings on teacher trainees’ health, which indicate that emotional exhaustion increases during the transition from university to the second phase of teacher education (referendariat) [82,83]. Only at the end of the referendariat was a decrease in emotional exhaustion indicated among teacher trainees [14,84,85]. The results of a study conducted during the COVID-19 pandemic replicated that teacher trainees were at a higher risk for high emotional exhaustion [86], which may have also influenced the results of our study.

The importance of *health literacy* for *well-being* can be deduced above all from the results of the multiple regression and correlations (Research Questions 3a and 3b). In terms of *health literacy*, the *self-regulation* dimension was the strongest predictor for *well-being,* with *self-control* and a *proactive approach to health* being additional predictors. Thus, Hypothesis 4 was only partially confirmed. Fiedler et al. [40] indicated similar associations between *health literacy* and *well-being* by industry managers. In their path model, *self-regulation* was the strongest predictor for *well-being*, whereas *proactive approach to health* had no direct effect on *well-being*, but ‘directly enables self-control and self-regulation at the ‘action-oriented’ level’ [40] (p. 7). In this context, our results should be further examined in future studies with latent and confirmatory designs. Nevertheless, Lenartz [20] was also able to explain over 40% of the variance in mental health. Without personal resource *health literacy*, the H, U, and A types had a significant association with *well-being* compared to the R type, but the variance clarification was small. If the *health literacy* dimensions were added, only the H type remained as a significant predictor for *well-being*, but the β-coefficient decreased. Overall, a stronger variance explanation was shown due to the dimensions of health literacy. Thus, Hypothesis 5 can be confirmed. An explanation for this could be derived from the post-hoc results. The H type had the highest means in all dimensions of *health literacy* and always differed significantly from the R type. Furthermore, statistical analyses can be used to confirm the assumption that the *self-regulation* and *self-control* dimensions of *health literacy* are related to *occupational self-regulation*.

## 7. Limitations

Although the results of our study add important findings to the field of health research for teacher trainees, this study still has some limitations that need to be highlighted. First, the data for the present study were collected online from teacher trainees in Germany. Thus, even though there was a variety in the distribution of age and school type, the findings may not generalise to teacher trainees outside Germany. Specifically, our findings may differ for countries with different educational systems. Second, the cross-sectional design did not allow causal inferences to be made between the investigated variables. In particular, a closer examination of the interplay between and causal direction of general health-related resources and in the context of the teaching profession should be followed in further studies for more insights. Third, we did not integrate other contextual factors or control variables into our analyses and therefore were not able to obtain a comprehensive picture of possible mediators and moderators of teacher trainees’ *general well-being*.

## 8. Conclusions

Overall, our study underlines that the *health literacy* dimensions self-regulation and self-control were mainly related to *occupational self-regulation*, and that the self-regulatory types and the health literacy dimensions self-regulation, self-control, and a proactive approach to health predicted teacher trainees’ *general well-being* independently of each other. These findings also have practical implications, as reflected by the magnitude of the effect sizes observed. First, teacher education and in-service teacher training that includes aspects of teachers’ *self-regulatory* skills and coping behaviour might enhance not only teacher trainees’ *well-being* but also their individual *health literacy*. Second, the *health literacy* of teacher trainees—especially the dimensions of *self-regulation* and *self-control*—are related to *occupational self-regulation*. However, the development of *occupational self-regulation* at an early stage through learning opportunities could also be associated with strengthening the personal resource *health literacy* of teacher trainees; therefore, the focus should be primarily on promoting emotional distancing and mental stability through intervention strategies.

## Figures and Tables

**Figure 1 ijerph-19-08821-f001:**
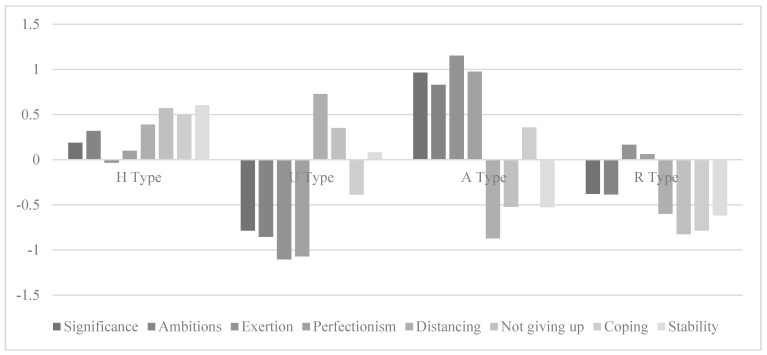
Z-scores for subscales of engagement and resilience by self-regulatory patterns. H = healthy-ambitious type; U = unambitious type; A = excessively ambitious type; R = resigned type.

**Table 1 ijerph-19-08821-t001:** Descriptive results for occupational self-regulation, health literacy, and well-being.

Scale	M	SD	α	N Items
**Occupational self-regulation** (scale range: 1–5) *Occupational engagement*				
Significance of work	2.36	0.79	0.85	4
Career ambitions	3.27	0.74	0.77	4
Exertion	3.10	0.89	0.83	4
Perfectionism	3.41	0.86	0.83	4
*Occupational resilience*				
Emotional distancing	2.85	0.84	0.85	4
Low tendency to give up	3.21	0.75	0.79	4
Active coping	3.33	0.67	0.82	4
Mental stability	3.35	0.68	0.71	4
**Health literacy** (scale range: 1–4)				
Self-regulation	2.45	0.56	0.78	5
Self-control	2.93	0.48	0.77	5
Self-perception	3.16	0.47	0.74	5
Proactive approach to health	2.96	0.51	0.82	5
Communication and cooperation	2.86	0.59	0.77	4
Dealing with health information	3.21	0.51	0.84	5
**Well-being** (scale range: 0–20)	13.13	5.00	0.85	5

*Note.**n* = 407, ***M*** Mean, ***SD*** Standard deviation.

**Table 2 ijerph-19-08821-t002:** Fit indices for different class solutions: Latent profile analysis for the class-dependent variance model.

Model	No. of Parameter	AIC	BIC	Sample-Adjusted BIC	Entropy	*p* LMR
1-class	16	9264.180	9328.321	9277.551	−	−
2-class	25	8847.867	8948.088	8868.759	0.735	0.0000
3-class	34	8675.911	8812.210	8704.323	0.735	0.0101
4-class	43	8558.533	8730.912	8594.466	0.756	0.6075
5-class	52	8494.500	8702.958	8537.955	0.776	0.0315

*Note.* AIC = Akaike information criterion; BIC = Bayesian information criterion; the Lo–Mendell–Ruben (LMR) test compares the current model to a model with *k*-1 profiles; *n* = 407.

**Table 3 ijerph-19-08821-t003:** Intercorrelations and correlations between occupational self-regulatory dimensions (1–8), health literacy dimensions (9–14), and well-being (15).

Scale			Self-Regulatory Dimensions								Health Literacy Dimensions					Well-Being
		1	2	3	4	5	6	7	8	9	10	11	12	13	14	15
1	Significance of work	−														
2	Career ambitions	0.53 **	−													
3	Exertion	0.38 **	0.40 **	−												
4	Perfectionism	0.34 **	0.43 **	0.62 **	−											
5	Emotional distancing	−0.24 **	−0.12 *	−0.57 **	−0.40 **	−										
6	Low tendency to give up	0.10 *	0.05	0.26 **	0.24 **	−0.49 **	−									
7	Active coping	0.26 **	0.29 **	0.16 **	0.21 **	0.14 **	0.46 **	−								
8	Mental stability	−0.01	0.07	−0.21 **	−0.06	0.37 **	0.46 **	0.25 **	−							
9	Self-regulation	−0.04	0.02	−0.43 **	−0.20 **	0.67 **	0.46 **	0.27 **	0.44 **	−						
10	Self-control	0.21 **	0.30 **	0.19 **	0.34 **	0.06	0.20 **	0.36 **	0.20 **	0.19 **	−					
11	Self-perception	0.02	0.12 *	−0.04	0.10 *	0.13 **	0.22 **	0.27 **	0.16 **	0.27 **	0.39 **	−				
12	Proactive approach to health	−0.04	0.08	−0.23 **	−0.06	0.23 **	0.18 **	0.18 **	0.11 *	0.34 **	0.26 **	0.42 **	−			
13	Communication and cooperation	−0.08	−0.04	−0.22*	−0.18 **	0.19 **	0.22 **	0.14 **	0.03	0.24 **	0.09	0.33 **	0.32 **	−		
14	Dealing with health information	−0.06	0.08	0.06	−0.03	0.09	0.16 **	0.17 **	0.17 **	0.11 *	0.25 **	0.42 **	0.26 **	0.14 **	−	
15	Well-being	0.07	0.15 **	−0.17 **	0.03	0.41 **	0.26 **	0.25 **	0.21 **	0.47 **	0.26 **	0.21 **	0.35 **	0.12 *	0.07	−

*Note.* * *p* < 0.05; ** *p* < 0.01.

**Table 4 ijerph-19-08821-t004:** Mean differences between the four self-regulatory patterns on health literacy: Results of analysis of variance (ANOVA).

Scale		Self-Regulatory Patterns				*ANOVA*		
		Type H *n* = 144	Type U *n* = 93	Type A *n* = 78	Type R *n* = 92	*F*	df	η^2^
Self-regulation	*M*	2.71	2.67	2.14	2.09	48.75 *	3	0.27
	*SD*	0.46	0.48	0.53	0.47			
Self-control	*M*	3.11	2.69	3.08	2.77	23.30 *	3	0.15
	*SD*	0.42	0.47	0.44	0.47			
Self-perception	*M*	3.25	3.11	3.20	3.01	5.31 *	3	0.04
	*SD*	0.41	0.52	0.44	0.48			
Proactive approach to health	*M*	3.07	3.00	2.86	2.82	5.85 *	3	0.04
	*SD*	0.51	0.50	0.51	0.51			
Communication and cooperation	*M*	2.95	3.02	2.68	2.69	8.74 *	3	0.06
	*SD*	0.59	0.57	0.56	0.57			
Dealing with health information	*M*	3.29	3.20	3.27	3.04	5.14 *	3	0.04
	*SD*	0.49	0.52	0.50	0.49			

*Note*. Means with unequal subscripts differ statistically significantly in the Turkey post-hoc tests. Type H = healthy-ambitious type; type U = unambitious type; type A = excessively ambitious type; type R = resigned type. * *p* < 0.0086.

**Table 5 ijerph-19-08821-t005:** Work-related and health-related personal resources and the association with general well-being: Results of the multiple regression.

	M_1_ β	M_2_ β	M_3_ β
Self-regulatory patterns			
H	**0.40 ****		**0.14 ***
U	**0.16 ***		−0.01
A	**0.12 ***		0.09
**Health literacy**			
Self-regulation		**0.39 ****	**0.38 ****
Self-control		**0.15 ****	**0.10 ***
Self-perception		0.00	0.00
Proactive approach to health		**0.21 ****	**0.21 ****
Communication and cooperation		−0.06	−0.05
Dealing with health information		−0.06	−0.07
R^2^	0.09	0.28	0.29

*Note.* R (resigned type) is the reference category. M = model; H = healthy-ambitious type; type U = unambitious type; type A = excessively ambitious type. For each regression, highly significant F value (*p* < 0.001), * *p* < 0.05, ** *p* < 0.001.

## Data Availability

The data that support the findings of this study are available from the corresponding author upon reasonable request.

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
