# Peer review of "Teacher Trainees’ Well-Being—The Role of Personal Resources"

_ijerph, 2022, doi:10.3390/ijerph19148821_

Round 1

Reviewer 1 Report

Thank you for the opportunity to review the submitted manuscript titled, “Teacher Trainees’ Well-Being -The Role of Personal and Occupational Resources.” This paper examines occupational self-regulation and health literacy as predictors of well-being among teacher trainees in Germany. Health literacy and occupational self-regulation present two overlapping constructs reflecting one’s initiative and motivation in meeting both their work- and health-related goals and needs. The examination of these two resources present an interesting contribution to the literature. The primary concern of this reviewer centers around the conceptualization of occupational self-regulation as an occupational resource. In theoretical models such as the Job Demands-Resources Model, occupational resources are more traditionally the byproduct of the work environment (e.g., effective communication pathways, supportive leadership, etc), while personal resources are typically resources from within the individual (e.g., personality, knowledge, internal motivation, etc.). Therefore, occupational self-regulation would typically be considered a personal resource, as it comes from within the individual, but rather a personal resource used to meet their occupational demands. This paper therefore seems to be comparing a work-related personal resource to a health-related personal resource in predicting teacher trainees’ well-being. Increased clarification in how these constructs align or don’t align with existing theories is important for placing this study within the organizational well-being literature. Below are a few additional pieces of feedback for consideration:

·     -There are numerous typos and grammatical errors throughout the manuscript. These should be addressed to promote clarity for readers.

·     - For readers less familiar with health literacy and occupational self-regulation, the authors would benefit from defining these constructs earlier in the introduction.

·    -The Job Demands-Resources Model is typically used to predict work-related well-being. On page two of the introduction, the authors reference the importance of also examining the positive dimensions of well-being. It’s important to note that the traditional application of the JDR model conceptualizes the positive aspects of well-being to still be work-related (e.g., work-related motivation and job satisfaction). This paper is examining well-being more broadly. The introduction would benefit from providing additional research in how work- and health-related personal resources such as health literacy and occupational self-regulation may or may not influence the general well-being of teachers.

·     -Throughout the paper the authors would benefit from making their focus on general well-being clearer (i.e., making it clear that they’re not examining work-related well-being).

Author Response

Dear Reviewer,

Thank you very much for your comments on our paper. We believe that incorporating your concerns has made our manuscript more structured and clearer.

The time and effort you put into your review is very much appreciated! We wrote a response to each one of your concerns and hope that we addressed all of your concerns adequately. Please let us know if you have any more improvements to suggest!

Sincerely,

Authors

Comment 1

The primary concern of this reviewer centers around the conceptualization of occupational self-regulation as an occupational resource. In theoretical models such as the Job Demands-Resources Model, occupational resources are more traditionally the byproduct of the work environment (e.g., effective communication pathways, supportive leadership, etc), while personal resources are typically resources from within the individual (e.g., personality, knowledge, internal motivation, etc.). Therefore, occupational self-regulation would typically be considered a personal resource, as it comes from within the individual, but rather a personal resource used to meet their occupational demands. This paper therefore seems to be comparing a work-related personal resource to a health-related personal resource in predicting teacher trainees’ well-being. Increased clarification in how these constructs align or don’t align with existing theories is important for placing this study within the organizational well-being literature.

Response 1

We thank you very much for your elaborations and thoughts and agree, that occupational self-regulation is indeed a personal resource used to meet occupational demands. Your point of view also aligns with the definition of Klusmann et al. (2012) who considers it a personal resource specific to the occupational context. We wanted to underline that occupational self-regulation is related to the work environment, by describing it as an occupational resource in our article. However, we now see, that this wording may be confusing. We therefore tried to clarify that we compare a work-related personal resource to a health-related personal resource in predicting teacher trainees’ well-being (as you pointed it out correctly). Moreover, we clarified how both constructs are aligned and believe that our paper benefits from this clarification.

Comment 2

There are numerous typos and grammatical errors throughout the manuscript. These should be addressed to promote clarity for readers.

Response 2

We did a proofreading, corrected all the type and grammar mistakes, and revised the manuscript in order to promote clarity for readers.

Comment 3

For readers less familiar with health literacy and occupational self-regulation, the authors would benefit from defining these constructs earlier in the introduction.

Response 3

We agree, that both constructs should be defined earlier in the introduction and modified our manuscript accordingly.

Comment 4

The Job Demands-Resources Model is typically used to predict work-related well-being. On page two of the introduction, the authors reference the importance of also examining the positive dimensions of well-being. It’s important to note that the traditional application of the JDR model conceptualizes the positive aspects of well-being to still be work-related (e.g., work-related motivation and job satisfaction). This paper is examining well-being more broadly. The introduction would benefit from providing additional research in how work- and health-related personal resources such as health literacy and occupational self-regulation may or may not influence the general well-being of teachers.

Response 4

We agree, that the Job Demands-Resources Model is typically used to predict work-related well-being and therefore deleted it form the Introduction. We now focus more on how the work- and health-related personal resources and their association with general well-being and clarified this in the introduction.

Comment 5

Throughout the paper the authors would benefit from making their focus on general well-being clearer (i.e., making it clear that they’re not examining work-related well-being).

Response 5

We thank you for this hint and modified our manuscript accordingly. In doing so we now point out in the Introduction that we focus in our study on teachers’ general well-being. Moreover, we refer to general well-being (instead of only well-being) throughout the paper.

We hope that our revisions make our paper more suitable for a publication in IJERPH. Thank you once again for your time and your valuable feedback!

Reviewer 2 Report

In general terms, the paper is well written and present all the information needed to understands the importance of the study.

1. Line 19, sentence “Latent profil analysis”. Do you mean profile with “e” at the end instead of profil?
INTRODUCTION
2. The authors did a good job presenting the lack of studies among student teachers on this topic and the theoretical ground of this analysis. RESEARCH QUESTIONS AND HYPOTHESES
3. The authors proposed several hypotheses to be proved. However, these hypotheses are well proposed METHOD
4. Participants are well described
5. Instruments are described sufficiently
6. The procedures description is short but clear
7. The statistical analysis is described in detailed
RESULTS
8. This section answers the research question in a proper way
9. The tables included support the text as expected
DISCUSSION
10. The discussion is well organized
11. In this section the authors try to explain the relationships found in the results
12. The limitations are acknowledged  

Author Response

Dear Reviewer,

Thank you very much for your appreciative feedback on our paper. We are more than happy that you enjoyed reading about this topic.

Sincerely,

Authors

Comment 1

Line 19, sentence “Latent profil analysis”. Do you mean profile with “e” at the end instead of profil?

Response 1

Thank you for pointing this out. We just forgot the "e" and have corrected it in the text.

We hope that our revisions make our paper more suitable for a publication in IJERPH. Thank you once again for your time and your valuable feedback!

Reviewer 3 Report

The first sentence of the Abstract reads funny. Seems like there should be a word after literacy (line 11)

Line 19 is profil a word? 

Line 53 One resource..... is occupational self-regulation. Is this a resource or a strategy or a construct?  I am not sure the word resource is the best term.

Section 5.2 is hard to follow. Just long text. Can this be broken up?   

 Line 169 only core dimensions... Should there be a figure?  

I think the conclusion section could be expanded based on all of the data presented. 

Author Response

Dear Reviewer,

We are very grateful that you provided us with valuable feedback on our work! We believe that incorporating your concerns has made our manuscript more structured and clearer.

Please let us know if you have any more improvements to suggest!

Sincerely,

Authors

Comment 1

The first sentence of the Abstract reads funny. Seems like there should be a word after literacy (line 11)

Response 1

We could understand your comment and rewrote the first sentences of the abstract and hope that it is now easier for the reader to understand. 

Comment 2

Line 19 is profil a word?

Response 2

We just forgot the "e" and have corrected it in the text. Thanks for this hint.

Comment 3

Line 53 One resource..... is occupational self-regulation. Is this a resource or a strategy or a construct?  I am not sure the word resource is the best term.

Response 3

We refer to the description by Klusmann et al. (2008) "[...] two work-related personal characteristics to foster an effective approach to occupational demands: work engagement and resilience" (p. 704). It is considered a personal resource specific to the occupational context (Klusmann et al., 2012), which is why we described it as an occupational resource in our article.  However, we now clarified that occupational self-regulation is indeed a personal resource used to meet occupational demands.

Comment 4

Section 5.2 is hard to follow. Just long text. Can this be broken up?  

Response 4

For a better overview and readability we have integrated several paragraphs in this chapter.

Comment 5

 Line 169 only core dimensions... Should there be a figure? 

Response 5

We have shortly described the structural model of health literacy, but in our paper, we are only referring to one part of the structural model, the core dimensions: self-perception, proactive approach to health, dealing with health information, self-control, self-regulation, communication and cooperation. We have deliberately decided against illustrating the whole structural model of health literacy in order not to confuse the reader and to focus exclusively on the core dimensions. Nevertheless, we can understand your comment and have referred to sources in which a figure of the model can be found.

Comment 6

I think the conclusion section could be expanded based on all of the data presented.

Response 6

We thank you for pointing this out and added some main points to the conclusion.

We hope that our revisions make our paper more suitable for a publication in IJERPH. Thank you once again for your time and your valuable feedback!